# Purposes of Internet Use and Its Impacts on Physical and Psychological Health of Korean Older Adults

**DOI:** 10.3390/healthcare12020244

**Published:** 2024-01-18

**Authors:** Gyeong-Suk Jeon, Kyungwon Choi

**Affiliations:** 1Department of Nursing, Division of Natural Science, Mokpo National University, Muan-gun 58554, Republic of Korea; sookie@mnu.ac.kr; 2Department of Nursing, Korea National University of Transportation, Chungbuk 27909, Republic of Korea

**Keywords:** internet use, older adults, self-rated health, depressive symptom

## Abstract

Older adults engage in online activities for various purposes. An increasing number of studies are establishing connections between the purposes of internet use and their impacts on health outcomes. This study aimed to examine how the purposes of internet use affect self-rated health and depressive symptoms among Korean older adults. A nationally representative survey of community-dwelling older adults in Korea was used in the analysis (*n* = 5094). Instrumental internet use (using the internet to access various forms of information, services, and other resources) in Korean older adults was as common as interpersonal communication use, and the purposes of internet use were mainly for information seeking for everyday needs and engaging in various activities for enjoyment. Internet use for interpersonal communication and instrumental purposes was significantly associated with depressive symptoms and self-rated health. Internet use for instrumental purposes had a greater impact on self-rated health (β = −0.186) and depressive symptoms (β = −0.160) compared with the impacts of interpersonal communication internet use on self-rated health (β = −0.101) and depressive symptoms (β = −0.047). The findings highlighted the necessity of designing interventions that take into account the different purposes of internet use for older adults in order to maximize its benefits, paying special emphasis for information seeking online.

## 1. Introduction

Rapid advances in information and communication technology have led to increasing internet use by older adults, such that engagement with online activities has become important in their everyday lives. This trend has been observed worldwide. For example, online use among older Americans aged ≥ 65 years increased from 28% in 2005 to 67% in 2016 [1]. In Europe, only 27% of people aged > 54 years used the internet in 2007; however, by 2016, 45% of people in this age group used the internet at least once per week, ranging from 14% in Greece to 80% in Sweden [2]. According to surveys in Korea, the internet use rate among older adults aged ≥ 60 years increased from 19.0% in 2008 [3] to 76.2% in 2022 [4], indicating steady growth during this period. Therefore, many gerontological studies have investigated the relationships between internet use and the well-being and health outcomes of older populations.

Studies examining the associations between internet use and the health outcomes of older adults have yielded inconsistent findings. Therefore, these associations remain inconclusive. Some studies have shown that internet use is associated with improvements in the well-being and various health outcomes of older populations. Using longitudinal data, Cotton et al. (2014) found that the likelihood of future depression was significantly lower (by 33%) among internet users than among non-users [5]. Similarly, Chopik [6] demonstrated that greater social technology use was significantly associated with better self-rated health, fewer chronic illnesses, higher subjective well-being, and fewer depressive symptoms. In studies of community-dwelling older adults, internet users reported higher levels of personal growth and purpose in life [7,8] and better self-rated health [7,9] than non-users; they also showed improved self-efficacy, self-control, and self-determination [10,11]. In contrast, some studies have shown no significant relationship between internet use and well-being or the life satisfaction of older adults, although there have been fewer such studies [12,13,14,15]. A rigorous randomized controlled trial by Slegers et al. [12] showed no effects of computer and internet use on cognitive function, autonomy, well-being, social network, or subjective physical functioning. A meta-analysis by Huang [14] showed that internet use had a slight negative impact on psychological well-being among older adults. These inconsistencies may be related to the simple comparison of the health outcomes between internet users and non-users in older populations, who likely have different purposes for internet use.

Older adults reportedly engage in online activities for various purposes, such as online communication to connect with others, leisure activities for entertainment, and the acquisition of health-related information [16,17,18,19]. Based on the data from community-dwelling older adults aged ≥ 65 years using the internet in the Netherlands, van Boekel et al. [20] identified four groups of individuals according to their purpose of internet use. Their study results showed that older adults are a diverse group with respect to activities on the internet. These findings highlight the importance of considering the purposes that older adults engage in online activities when examining the relationship between internet use and well-being among older populations [21]. Indeed, in a study of Canadian community-dwelling older adults aged ≥ 60 years, internet use for communication and information-seeking purposes was positively associated with life satisfaction, self-efficacy, and social support and negatively associated with depression. However, there were no significant associations with internet use for entertainment and health purposes [11]. A study focusing on the oldest old (individuals aged ≥ 80 years) in the United States showed that social motivations played a mediating role in the relationships between information and communication technology use and life satisfaction. However, informational motivations mediated the relationships of information and communication technology use with physical and subjective health, but not life satisfaction [22]. Sum et al. [23] reported that among older adults, internet use for communication (rather than information-seeking purposes) was linked to reduced loneliness, whereas instrumental uses such as commerce and entertainment showed no significant association with well-being.

In summary, an increasing number of studies are establishing connections between the purposes of internet use and their impacts on health outcomes. Nevertheless, the gap identified from the literature review is the lack of sufficient research in this domain. Such research is often hindered by small sample sizes, resulting in limited empirical analyses. In many governmental regulations in South Korea, internet infrastructure has the highest priority; thus, South Korea has one of the fastest and most extensive internet networks worldwide. Considering these findings along with the widespread availability of high-speed internet, it is unsurprising that 93% of South Koreans aged ≥ 3 years are internet users. Indeed, a recent study showed that the internet use rate among older adults aged 60–69 years was 94.0% [4]. Furthermore, among the member countries of the Organization for Economic Co-operation and Development, the rate of internet use by adults aged 55–74 years in South Korea (87.4%) was the seventh highest worldwide [24]. In this context, it is necessary to investigate the differential impact of the purposes of internet use on physical and psychological health among older adults in Korea with high-speed internet and high rates of internet use. Therefore, the present study aimed to examine how various purposes of internet use affect self-rated health and depressive symptoms among a representative sample of older adults in Korea.

## 2. Materials and Methods

### 2.1. Data and Participants

The data were derived from the 2020 Living Profiles of Older People Survey (LPOPS). The LPOPS is conducted every 3 years by the Korean Ministry of Health and Welfare. The study was officially approved by the Institutional Review Board of the Korea Institute for Health and Social Affairs (KIHSA IRB Number: 2020-36, approved on 15 June 2020).The 2020 LPOPS involved a cross-sectional and nationally representative sample of community-dwelling older adults (aged ≥ 65 years) who lived within 17 regions (seven metropolitan cities, one special self-governing city, and nine provincial regions) in both urban and rural areas of South Korea. A nationwide probability sample of non-institutionalized older adults was selected using a stratified two-stage cluster sample design. In total, 10,097 adults aged ≥ 65 years participated in the survey. After informed consent had been obtained, trained interviewers made house-to-house visits and conducted face-to-face interviews with the participants using a tablet personal computer (PC)-assisted personal interview technique from 14 September to 20 November 2020. Additional details of the survey design and methods have been reported elsewhere [25]. We excluded participants who responded by proxy and participants with missing data relevant to this analysis (i.e., depressive symptoms and income). Finally, 2338 men and 2756 women aged ≥ 65 years (mean age 70.71 ± 4.97 years) who used internet devices (tablet PCs, smartphones, or computers) were included in the analysis.

### 2.2. Depressive Symptoms and Self-Rated Health

Depressive symptoms were evaluated using the Short-form Geriatric Depression Scale–Korean version (SGDS-K). The Geriatric Depression Scale (GDS) was developed by Yesavage and Sheik [26] and translated into South Korean by Bae and Cho [27]. Compared with the 30-item K-GDS, the 15-item SGDS-K (Short-form Geriatric Depression Scale–Korean version) exhibited equal validity and was less affected by education, current employment, living accommodations, and chronic diseases [28]. Its 15 items included five positive and 10 negative feelings for the previous week, where each was scored as negative (1) or positive (0); the total score ranged from 0 to 15. The five items with positive feelings were inverted, and higher summed scores indicated more severe depression. The SGDS-K has demonstrated high reliability (Cronbach’s alpha of 0.90) and validity [28], and Cronbach’s α coefficient in the present study was 0.809. We also measured global self-rated health using the question “How would you rate your health in general?” The five response options for this question were as follows: 1 = very good, 2 = good, 3 = fair, 4 = poor, and 5 = very poor. The total score ranged from 1 to 5, and higher scores indicated worse health.

### 2.3. Purpose of Internet Use

According to the perceived usefulness of certain purposes of internet use, the participants’ internet use was divided into two categories: interpersonal communication and instrumental use. Each participant was presented with examples of 11 types of internet use involving a PC, smartphone, or tablet PC and asked to respond to all the examples. Internet use for interpersonal communication was scored 1 if the respondent used the internet for any type of interpersonal communication. The same criterion was applied to instrumental use. Interpersonal communication was defined as using the internet to communicate with others and included chatting by voice, video, and text. Internet use for interpersonal communication was measured with three items: sending text messages using various social networking service applications such as Telegram and Kakao Talk; receiving text messages using such applications; and engaging in various social networking service activities such as Facebook, Instagram, and Twitter. Instrumental use was defined as using the internet to access various forms of information, services, and other resources and was measured with eight items: information searching and inquiry; photographing or filming; listening to music; playing games; watching movies, YouTube videos, or television programs; internet commerce (e.g., shopping and/or booking transportation); financial activities (e.g., banking and/or stock trading); and searching for and installing applications. In total, 4853 (95.8%) older adults used the internet for interpersonal communication and 4415 (86.7%) used it for instrumental purposes.

### 2.4. Covariates

The covariates in this study were age (65–74, 75–84, and ≥85 years), sex, marital status (married, widowed, or other), residence (urban or rural), living with children (yes or no), education, equivalent annual household income, economic activity (yes, no, or never), and time spent using internet devices each week. The equivalent annual household income (total household income divided by the square root of the number of household members) was calculated and divided into quartiles to detect a nonlinear relationship. The time spent using internet devices each week was calculated according to the time per day and days per week using a PC, smartphone, or tablet PC, excluding phone calls and text messages reported by each participant.

### 2.5. Statistical Analysis

The data were expressed as frequencies, weighted proportions, and means ± standard deviations of the baseline characteristics according to sex. The chi-square test and *t*-test were performed to compare the distributions of the frequencies and means between men and women. We also descriptively examined the percentage distributions of internet use for interpersonal communication and instrumental purposes according to each variable by sex. A multiple linear regression analysis was conducted to assess the impact of internet use for interpersonal communication and instrumental purposes on depressive symptoms and self-rated health. We evaluated the possible multicollinearity between the covariates, such as education and equivalent household income, via a correlation analysis and collinearity statistics (tolerance and variance inflation factor tests). No significant collinearity was detected between any of the covariates. All the statistical analyses were conducted using SPSS version 27.0 for Windows (IBM Corp., Armonk, NY, USA).

## 3. Results

Table 1 shows the descriptive statistics of the study sample. The proportions were weighted according to the sample design. Among the study participants, the mean ages of women (70.19 years) and men (71.26 years) were similar. In rural areas, older men (18.9%) outnumbered older women (16.7%). A total of 31.3% of the women were widows, whereas only 8.0% of the men were widowers. The percentage of participants with an elementary school education or no education was higher among older women (29.5%) than among older men (14.7%), and more older men (58.1%) than older women (38.5%) participated in economic activity. Most of the older men and women were living with children (85.2% and 79.1%, respectively). The mean SGDS-K score was higher for women (3.08 ± 3.03) than for men (2.63 ± 2.85), and the incidence of depression was higher among women (4.3%) than among men (2.8%). More than half of the older men and women self-reported their health as good (66.1% and 57.7%, respectively). The mean time spent using internet devices each week was higher among men (10.79 ± 11.33 h) than among women (8.60 ± 9.52 h). Most older men and women reported internet use for interpersonal communication (96.1% and 95.1%, respectively); 89.8% of men and 85.6% of women reported internet use for instrumental purposes. With respect to interpersonal communication, both men and women received more messages (95.8% and 94.2%, respectively) than they sent (91.1% and 88.3%, respectively). The most prevalent purpose of instrumental internet use in older men and women was information searching/inquiry (80.7% and 69.5%, respectively), followed by photographing or filming (76.6% and 72.0%, respectively), and then watching movies, YouTube videos, and television programs (59.8% and 51.6%, respectively).

The mean scores of internet use for both interpersonal communication and instrumental purposes were higher among men than among women. Overall, the mean scores of internet use for interpersonal communication and instrumental purposes were higher among younger individuals, individuals who were married, individuals who practiced religion, and individuals who lived in urban areas. A higher education level, higher equivalent household income, and participation in economic activity were associated with higher internet use among older adults. Older adults who had no depressive symptoms and self-reported their health as good exhibited more frequent internet use for both interpersonal communication and instrumental purposes, compared with their counterparts (Table 2).

Table 3 shows the results of the multiple linear regression analysis conducted to assess the impact of interpersonal communication (Model 1) and instrumental use (Model 2) of internet devices on self-rated health and depressive symptoms after controlling for covariates (sex, age, marital status, living with children, religion, residency area, education, equivalent household income, and economic activity). The utilization of Models 1 and 2 to assess subjective health showed that internet uses for interpersonal communication (β = −0.101) and instrumental purposes (β = −0.186) were associated with self-rated health. With respect to depressive symptoms, internet uses for interpersonal communication (β = −0.047) and instrumental purposes (β = −0.160) were also associated with depressive symptoms in Models 1 and 2.

## 4. Discussion

This study was performed to explore the impact of internet use on physical and psychological health among older populations by stratifying the purposes of online activities. We found that internet use for interpersonal communication and instrumental purposes in older adults was significantly associated with a higher level of self-rated health and lower scores of depressive symptoms. Moreover, unlike previous studies, our study showed that internet use for instrumental purposes had a greater positive impact on both physical health and psychological well-being, compared with internet use for interpersonal communication. In previous studies, internet use for interpersonal communication was associated with higher levels of life satisfaction and lower levels of depression and social loneliness [11,21,22,23], whereas internet use for instrumental purposes was associated with worse physical and subjective health [22], increased levels of social loneliness, and more severe depressive symptoms [29,30]. These inconsistencies relative to the previous studies may reflect different values of internet use for instrumental purposes between older adults in Korea and older adults in Western countries, where most of the previous studies were conducted. According to a report on the digital divide in Korea [31], middle-aged and older adults (aged ≥ 55 years), as well as most Korean respondents, used digital devices with similar frequencies for information searching (76.8% and 86.2%, respectively) and interpersonal communication (76.7% and 85.0%, respectively). Moreover, in longitudinal studies concerning the effects of internet use among older populations in Korea, internet use by older adults had significant effects on life satisfaction [32] and depressive symptoms [33]. Internet use for information and services among Korean older adults was significantly associated with higher levels of self-esteem and life satisfaction and lower levels of depressive symptoms and anxiety, whereas internet use for interpersonal communication did not show these associations [34]. Therefore, it is unsurprising that internet use for instrumental purposes and interpersonal communication among Korean older adults was positively associated with both higher levels of self-rated health and lower levels of depressive symptoms.

The positive relationships between instrumental internet use with higher levels of self-rated health and lower levels of depressive symptoms also corresponded to the higher rates and more varied types of instrumental internet use in the present study than in studies from other countries. In this study, internet activities performed by more than half of the participants were sending and receiving messages (89.6% and 94.9% respectively); searching for information (74.6%); taking photographs or making videos (74.6%); and watching movies, YouTube videos, or television programs (55.4%). However, in a study of 1211 community-dwelling older adults (aged ≥ 65 years) in England [35], the percentages of internet use activities were much lower than the percentages in the present study, where the most prevalent types of internet use were searching for information (49.2%), writing e-mails (37.5%), and viewing pictures and videos (30.7%). In another study using a large U.K. sample of 3500 middle-aged and older adults (aged 55–75 years) from the English Longitudinal Study of Ageing cohort study [30], e-mail communication was the most popular reason for internet use (80%), followed by shopping for and buying goods (68%), making videos or voice calls (58%), managing finances (56%), and reading the news (52%). Overall, unlike the findings in the previous studies, instrumental internet use in Korean older adults was as common as interpersonal communication use, and the purposes were mainly for information seeking for everyday needs and engaging in various activities for enjoyment rather than daily living services (i.e., the most common reason for use in the two U.K. studies). These results were consistent with reports that most digital device use by Korean older adults was for instrumental purposes, such as information searching and engaging in activities for enjoyment (games, movies, and music) rather than interpersonal communication [36,37]. In an analysis of the longitudinal trends (2004–2014) concerning the use and non-use of information technology by older adults, Kim and Jeon [38] reported that the main goals of internet use by Korean older adults had changed; there were increases in information seeking, watching movies and listening to music, and taking classes, along with decreases in e-mailing and news consumption.

A possible explanation for the higher value of instrumental internet use by Korean older adults may be the rapid change in the social value of living situations among older adults and the fast internet speed with nationwide broadband infrastructures in South Korea. Frist, rapid social changes, accompanied by the modernization and urbanization of Korean society, have led to a decline in intergenerational co-residence and an increase in the proportion of older adults living alone or with a spouse (from 40.4% in 1990 to 83.7% in 2020) [25,39]. Moreover, the percentage of older adults who did not want to live with their adult children increased from 72.4% in 2011 [40] to 87.2% in 2020 [25]. This trend has been strengthened since the Korean baby boomer generation, which represents a substantial segment of the older demographic, entered the ranks of older adults beginning in the year 2020. This generation comprises approximately 7.13 million individuals born from 1955 to 1963 after the Korean War. In the present study, older adults aged 65–69 years comprised more than half of the population (51.2%) and had the highest scores for instrumental internet use among all age groups, which may contribute to this phenomenon. This generation, often referred to as the “sandwich” generation because they are simultaneously responsible for the care of a child and an older adult (in contrast to the previous generation, which typically relied on their adult children for caregiving in old age), tends to prioritize independence and place relatively high emphasis on the importance of a spouse [41,42]. The perceptions of aging held by these baby boomers were not correlated with age-based stereotyping and discrimination against older individuals [43]. Baby boomers have defined successful aging as the culmination of several life factors, including health, leisure, finances, volunteer work, family relationships, and the ability to live independently in old age without relying on their children for assistance [44]. As they gain familiarity with information technologies such as smartphones, boomers rapidly adapt to technological advancements, and they actively use the internet for various purposes [45,46]. As aspiring learners, they have a strong desire to understand how to use information technologies to create new hobbies or expand their enjoyment in older age [47,48]. In conclusion, with the shift in perspective concerning life in old age and the entry of baby boomers into the older population, Korean older adults may place greater emphasis on the access to and acquisition of diverse information in their everyday lives.

Second, South Korea has one of the world’s most rapid internet speeds and the highest rate of internet use (96%) [49,50]. Since the introduction of the first nationwide fifth-generation (5G) services in 2019, the country now has one of the world’s most extensive broadband infrastructures and the fastest internet connection speeds [51]. Paired with the widespread availability of high-speed internet, the internet is considered more important for many Koreans than for individuals in the West. According to a 2022 report on the digital divide [52], 98.3% of Koreans have a smartphone and 93.0% of Koreans access the internet in their daily lives. According to a 2019 internet usage survey in Korea [53], many Koreans aged > 6 years engage in online activities on a daily basis, such as searching for various kinds of information (96.5%), sending and receiving e-mails (61.7%), performing internet banking (64.9%) (people aged > 12 years), buying goods online (or engaging in e-commerce) (64.1%), and using online media (96.5%). In South Korea, numerous traditional offline jobs can now be performed online through mobile apps or websites, often without requiring direct interpersonal communication. This functionality has been extended to government services through the digitalization of social services. Moreover, the COVID-19 pandemic made social distancing a necessity, leading to the adoption of “untact” practices as the new norm in daily life. These practices include online food ordering, self-service kiosks, online shopping, and contactless payments. According to a 2002 internet usage survey in Korea [4], internet use in people’s daily lives has increased since 2019 [53]. For example, from 2019 to 2022, internet shopping increased from 64.1% to 73.7%, internet banking increased from 64.9% to 79.2%, internet classes increased from 24.6% to 39.8%, social network use increased from 63.8% to 69.7%, and booking transportation tickets increased from 44.8% to 53.5% [54]. The use of automated ordering such as self-service kiosks among older adults aged ≥ 60 years also increased from 39.2% in 2021 [54] to 71.4% in 2022 [4]. Therefore, it has become essential to have access to internet services and a comprehensive understanding of various internet uses to adapt to Korea’s post-COVID-19 and digitally oriented society, particularly for older adults who seek to maintain autonomy and independence in their daily lives.

This study had some limitations. First, its cross-sectional design hindered inferences regarding causality or exploring a bidirectional relationship between the purpose of internet use and self-rated health/depressive symptoms among older adults. A study with a longitudinal design is needed to fully understand the directions of the effects and causality of the relationships identified in the present work. Second, the influence of the COVID-19 pandemic should be considered when interpreting the study findings because we used data from the 2020 LPOPS, which was conducted during the COVID-19 pandemic. The pandemic may have caused distress in older adults and increased their amount of time spent online as well as their need for information, thus affecting their psychological status and the type and frequency of their internet use. Third, only one item was utilized to measure each type of internet use. The participants were only asked whether they used the internet for a specific purpose and the total amount of time spent online, rather than how much time they spent on each type of internet use or how frequently they engaged in particular types of internet use. Despite these limitations, our study used a nationally representative sample weighted by census estimates, thereby increasing the generalizability of the findings. Furthermore, because this investigation was conducted in Korea, where high-speed internet is readily accessible and most older adults use the internet in their daily lives [4], we believe that the effects of using the internet on older adults’ physical and psychological health were more clearly demonstrated in this study than in studies performed in countries with limited internet access.

## 5. Conclusions

The results of this study emphasized the significance of internet use for instrumental purposes and interpersonal communication in relation to the physical and psychological well-being of older adults in Korea. Specifically, the adverse relationship of instrumental internet use with depressive symptoms contradicted the previous findings [29,30] and provides new insight into the influence of internet use on psychological health among older adults in South Korea. Overall, the findings of this study highlight the need for interventions that consider the various types of internet use to maximize their benefits among older adults, with particular attention toward internet use for information seeking.

## Figures and Tables

**Table 1 healthcare-12-00244-t001:** Sociodemographic characteristics, health status, and internet use characteristics of Korean older adults using the internet.

	Men		Women		*p*	Total	
	*n*	%	*n*	%		*n*	%
*N*	2338	45.9%	2756	54.1%		5094	100.0%
Age (years)	71.26	4.98	70.19	4.90	<0.001	70.71	4.97
65–69	1179	45.1%	1620	56.9%	0.034	2799	51.2%
70–74	698	29.6%	668	22.7%		1366	26.0%
75–79	317	18.1%	314	14.6%		631	16.3%
≥80	144	7.2%	154	5.9%		298	6.5%
Area of residence							
Urban	1805	81.1%	2203	83.3%	<0.001	4008	82.2%
Rural	533	18.9%	553	16.7%		1086	17.8%
Religion							
Yes	1022	42.7%	867	31.7%	<0.001	1889	37.0%
No	1316	57.3%	1889	68.3%		3205	63.0%
Marital status							
Married	1937	88.9%	1533	64.6%	<0.001	3470	76.3%
Widowed	273	8.0%	1063	31.3%		1336	20.1%
Others	128	3.0%	160	4.2%		288	3.6%
Education (years)	10.92	0.03	9.29	0.03	<0.001	10.08	0.03
College or beyond	321	15.1%	109	4.6%	<0.001	430	9.7%
High school	1113	48.2%	969	37.4%		2082	42.6%
Middle school	547	22.0%	801	28.4%		1348	25.3%
Elementary school	324	13.3%	750	24.9%		1074	19.3%
Uneducated	33	1.4%	127	4.6%		160	3.1%
Equivalent household income ^a^	2588.33	3024.04	208.52	3950.17	0.038	2547.00	3533.99
Highest 25%	895	38.7%	879	35.3%	<0.001	1774	36.9%
Second 25%	704	30.8%	738	26.8%		1442	28.7%
Third 25%	402	15.9%	596	19.7%		998	17.9%
Lowest 25%	337	14.7%	543	18.2%		880	16.5%
Economic activity							
Yes	1358	58.1%	1062	38.5%	<0.001	2420	47.5%
No	942	40.3%	1187	43.1%		2129	41.8%
Never	38	1.6%	507	18.4%		545	10.7%
Living with children							
Yes	1987	85.2%	2298	79.1%	0.064	4285	82.1%
No	351	14.8%	458	20.9%		809	17.9%
Depressive symptoms ^b^ (SGDS-K score)	2.63	2.85	3.08	3.03	<0.001	2.86	2.95
None (<5)	1807	77.6%	2026	72.9%	<0.001	3833	75.2%
Mild (5–9.99)	454	19.7%	608	22.8%		1062	21.3%
Depression (≥10)	77	2.8%	122	4.3%		199	3.5%
Self-rated health (1–5)	2.35	0.795	2.53	0.817	<0.001	2.44	0.811
Healthy	1555	66.1%	1596	57.7%	<0.001	3151	61.7%
Fair	569	24.6%	798	28.5%		1367	26.6%
Unhealthy	214	9.3%	362	13.8%		576	11.7%
Information technology use per week (hours)	10.79	11.33	8.60	9.52	<0.001	9.65	10.49
Interpersonal communication (0–3)	2.29	0.73	2.18	0.75	<0.001	2.24	0.74
Yes	2247	96.1%	2606	95.1%	<0.001	4853	95.6%
Receiving messages (Telegram, Kakao Talk, or others)	2240	95.8%	2596	94.2%		4836	94.9%
Sending messages (Telegram, Kakao Talk, or others)	2129	91.1%	2433	88.3%		4562	89.6%
Social networking activities (Facebook, Instagram, Twitter, or others)	992	42.4%	956	34.7%		1948	38.2%
Instrumental use (0–8)	3.53	2.29	2.94	2.12	<0.001	3.22	2.22
Yes	2082	89.8%	2333	85.6%		4415	87.6%
Information searching and inquiry	1886	80.7%	1915	69.5%		3801	74.6%
Photographing or filming	1791	76.6%	1984	72.0%		3775	74.1%
Listening to music (MP3, radio, etc.)	1068	45.7%	1076	39.0%		2144	42.1%
Playing games	672	28.7%	622	22.6%		1294	25.4%
Watching movies, YouTube videos, television programs, etc.	1398	59.8%	1422	51.6%		2820	55.4%
Internet commerce (shopping, booking, etc.)	356	15.2%	287	10.4%		643	12.6%
Financial activities (banking, stock trading, etc.)	569	24.3%	424	15.4%		993	19.5%
Application use	478	20.4%	317	11.5%		795	15.6%

The data are presented as *n* with a weighted % or mean with standard deviation. ^a^ The equivalent household income was adjusted for the number of people in the household and divided into quartiles. ^b^ Depressive symptoms were measured by scores on the 15-item Short-form Geriatric Depression Scale–Korean version (SGDS-K).

**Table 2 healthcare-12-00244-t002:** Purposes of internet use according to the sociodemographic and health characteristics among Korean older adults using the internet.

	*n*	%	Interpersonal Communication			Instrumental Use		
			Mean	SD	*p*	Mean	SD	*p*
Total	5094		2.24	0.74		3.22	2.22	
Sex					<0.001			<0.001
Male	2338	45.9%	2.29	0.73		3.53	2.29	
Female	2756	54.1%	2.18	0.75		2.94	2.12	
Age (years)					<0.001			<0.001
65–69	2799	51.2%	2.41	0.61		3.83	2.19	
70–74	1366	26.0%	2.21	0.72		2.94	2.06	
75–79	631	16.3%	1.93	0.85		2.32	2.01	
≥80	298	6.5%	1.71	1.00		1.82	1.95	
Marital status					<0.001			<0.001
Married	3470	76.3%	2.29	0.71		3.37	2.23	
Widowed	1336	20.1%	2.01	0.85		2.52	2.04	
Other	288	3.6%	2.44	0.64		3.95	2.12	
Religion					<0.001			<0.001
Yes	1889	37.0%	2.28	0.69		3.35	2.17	
No	3205	63.0%	2.21	0.77		3.15	2.25	
Area of residence					<0.001			<0.001
Urban	4008	82.2%	2.26	0.72		3.29	2.21	
Rural	1086	17.8%	2.13	0.82		2.92	2.23	
Education					<0.001			<0.001
College or beyond	430	9.7%	2.62	0.58		4.89	2.47	
High school	2082	42.6%	2.40	0.60		3.73	2.07	
Middle school	1348	25.3%	2.20	0.69		2.86	1.95	
Elementary school	1074	19.3%	1.87	0.87		2.13	1.89	
Uneducated	160	3.1%	1.31	0.92		0.88	1.30	
Equivalent household income ^a^					<0.001			<0.001
Highest 25%	1774	36.9%	2.40	0.66		3.86	2.25	
Second 25%	1442	28.7%	2.25	0.70		3.04	2.00	
Third 25%	998	17.9%	1.98	0.82		2.43	2.06	
Lowest 25%	880	16.5%	2.11	0.81		3.00	2.32	
Economic activity					<0.001			<0.001
Yes	2420	47.5%	2.32	0.71		3.59	2.26	
No	2129	41.8%	2.18	0.76		3.00	2.17	
Never	545	10.7%	2.10	0.79		2.58	2.02	
Living with children					0.049			0.070
Yes	4285	82.1%	2.23	0.74		3.20	2.22	
No	809	17.9%	2.25	0.77		3.34	2.23	
Depressive symptoms ^b^ (SGDS-K score)					<0.001			<0.001
None (<5)	3833	75.2%	2.28	0.73		3.45	2.25	
Mild (5–9.99)	1062	21.3%	2.10	0.75		2.56	1.96	
Depression (≥10)	199	3.5%	2.07	0.85		2.40	2.01	
Self-rated health					<0.001			<0.001
Healthy	3151	61.7%	2.36	0.70		3.70	2.21	
Fair	1367	26.6%	2.09	0.72		2.60	1.96	
Unhealthy	576	11.7%	1.89	0.87		2.10	2.06	

SD, standard deviation. ^a^ The equivalent household income was adjusted for the number of people in the household and divided into quartiles. ^b^ Depressive symptoms were measured by scores on the 15-item Short-form Geriatric Depression Scale–Korean version (SGDS-K).

**Table 3 healthcare-12-00244-t003:** Multiple linear regression for self-rated health and depressive symptoms among Korean older adults using the internet for interpersonal communication and instrumental purposes.

	Self-Rated Health		Depressive Symptoms ^b^	
	Model 1	Model 2	Model 1	Model 2
Interpersonal communication	−0.101 ***		−0.047 **	
Instrumental use		−0.186 ***		−0.160 ***
Sex	0.045 **	0.036 *	0.022	0.014
Age	0.181 ***	0.164 ***	0.060 ***	0.037 *
Married	−0.019	−0.018	−0.075 ***	−0.072 ***
Living with children	0.017	0.022	−0.032 *	−0.027 *
Religion	0.056 ***	0.059 ***	0.002	0.005
Rural resident	0.022	0.018	0.025	0.020
Education years	−0.094 ***	−0.078 ***	−0.107 ***	−0.084 ***
Equivalent household income ^a^	−0.055 ***	−0.048 ***	−0.054 ***	−0.047 ***
Economic activity	0.138 ***	0.131 ***	0.043 **	0.036 **
Information technology use per week (hours)	−0.036 **	0.005	0.017	0.060 ***
*R* ^2^	0.155	0.171	0.052	0.068

^a^ The equivalent household income was adjusted for the number of people in the household and divided into quartiles. ^b^ Depressive symptoms were measured by scores on the 15-item Short-form Geriatric Depression Scale–Korean version (SGDS-K). * *p* < 0.05, ** *p* < 0.01, *** *p* < 0.001.

## Data Availability

Data are contained within the article.

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
