# Peer review of "Purposes of Internet Use and Its Impacts on Physical and Psychological Health of Korean Older Adults"

_healthcare, 2024, doi:10.3390/healthcare12020244_

Round 1

Reviewer 1 Report

Comments and Suggestions for Authors

There are some minor terminological and editorial issues which need to be addressed. Please find attached file for improvement suggestions.

Author Response

Comment #1. Please, include SRH (self-rated health) in the abstract.

â–ºResponse: According to the reviewer’s comment, we rephrased self-rated health instead of acronym, SRH in abstract.

Comment #2. Revise the numerical notation in the following sentence “Finally, 2338 men and 2756 women aged ≥ 65 years (mean age, 70.71 ± 4.97 years) who used internet devices (tablet PCs, smartphones, or computers) were included in the analysis”.

â–ºResponse: We have put the numerical notations in the sentence (Line 113).

Comment #3. Please, format statistical analysis, as 2.4 and the other subchapters

â–ºResponse: We formatted statistical analysis as subchapter 2.5 (Line 166)

Comment #4. Explain Equivalent household income (10,000 won) in Table 2 as the equivalent in USD and whether this is average, below average, mean, and well off, etc.

â–ºResponse: We deleted Korean monetary unit, and revised equivalent household income groups as highest 25%, second 25%, third 25%, and lowest 25% (Table 1, 2)

Comment #5. In Discussion, revise Youtube as YouTube videos.

â–ºResponse: We have corrected the term as the reviewer’s comment (Line 149, 201, 281, Table 1)

Reviewer 2 Report

Comments and Suggestions for Authors

Thank you for the opportunity to review this paper. Overall, the authors did a good job articulating the research gaps and how this paper adds to existing knowledge. Minor comments could improve the work and, hopefully, the overall paper. 

1. In the abstract - please spell out the acronym SRH and use South Korea consistently throughout the entire manuscript. I would also suggest that SRH is spelt out in full throughout the manuscript, as it was difficult having to scroll up to find its meaning. Likewise, the meaning of instrumental purposes should be briefly stated in the abstract for better clarity.

2. Under the purpose of internet use (2.3), could you provide a source for the definition of instrumental and interpersonal communication use?

3. Under results, expressions like 'more older men' -line 182, need to be quantified or specified to reflect which age group. Likewise, acronyms such as SGDS-K should also be spelt out for clarity. There are also some types in the table, such as incomea, and symptomsb. 

4. Conclusion - Could you please support the claim that self-rated health is a measure of physical health with evidence?

Comments on the Quality of English Language

There are many complex sentences that could be simplified by using full stops as opposed to semi-colon. An example is the sentence from lines 60 to 64; 85 to 88; 

Author Response

Comment #1. In the abstract - please spell out the acronym SRH and use South Korea consistently throughout the entire manuscript. I would also suggest that SRH is spelt out in full throughout the manuscript, as it was difficult having to scroll up to find its meaning.

â–ºResponse: We rephrased self-rated health instead of acronym, SRH throughout the manuscript.

Comment #2. Likewise, the meaning of instrumental purposes should be briefly stated in the abstract for better clarity.

â–ºResponse: According to the reviewer’s comment, we added the meaning of instrumental internet use in the abstract (Line 14-15).

Comment #3. Under the purpose of internet use (2.3), could you provide a source for the definition of instrumental and interpersonal communication use?

â–ºResponse: In literature, types of internet use are generally categorized into three or four; informational, instrumental/ entertainment, social use [1]. However, instrumental uses of ICTs indicate using ICTs as convenient means of ‘instrument’ to obtain useful information, services, or other resources without direct interaction with others [2]. Ihm & Hsieh [3] also classified types of internet use of older adults into two categories (social and instrumental) to address different impacts of various uses of ICTs among older adults on their social engagement and quality of life. The aim of this study was to investigate the impacts of internet use for social network maintenance and non-communicational use on the physical and psychological health of older adults. Based on previous studies [2,3], We divided the purposes of diverse online activities into two categories: interpersonal communication and instrumental usage in daily life (including information searching and entertainment). However, as the reviewer’s concern, we now think the meaning of the term, “the instrumental internet use” used in this study may be misleading because it covers all usages of internet except social use unlike in previous studies. To make the aim of this study clearer, we have changed the title of this manuscript to read “Purposes of internet use and its impacts on physical and psychological health of Korean older adults”.

  1. Szabo A, Allen J, StephensC, Alpass (2019). Longitudinal analysis of the relationship between purposes of internet use and well-being among older adults. Gerontologist, 59(1), 58-68.
  2. Campbell SW, Kwak N. (2010). Mobile communication and civic life: Linking patterns of use to civic and political engagement. Journal of Communication, 60(3), 536–555.
  3. Ihm J, Hsieh YP. (2015). The implications of information and communication technology use for the social well-being of older adults. Information, communication & Society, 18(10). 1123-1138.

Comment #3. Under results, expressions like 'more older men' -line 182, need to be quantified or specified to reflect which age group.

â–ºResponse: We revised that sentence as the reviewer suggested (Line 183-184).

Comment #4. Likewise, acronyms such as SGDS-K should also be spelt out for clarity. There are also some types in the table, such as incomea, and symptomsb. 

â–ºResponse: As for SGDS-K, we inserted the full form of the term SGDS-K in manuscript (Line 118-119, 121-122), and in every Table footnotes, and revised the phrases with superscript.

Comment #5. Conclusion - Could you please support the claim that self-rated health is a measure of physical health with evidence? 

â–ºResponse: Self-rated health, a consistent predictor of mortality [1, 2] and morbidity [3], is a widely used measure of general health status. Over objective health measures, SRH may signify an individual's superior understanding of their own health status and health risks [4]. Poor SRH is associated with faster decline in functional ability of older adults [5,6], and the association of poor SRH with mortality has been reported to be almost as strong as that of poor objective health status [7]. SRH is statistically, not causally, linked to a decline in physical function [4]. Based on these, we presumed in this study that Self-rated health represents general physical health status of older adults.

  1. Idler EL, Benyamini Y. (1997). Self-Rated Health and Mortality: A Review of Twenty-Seven Community Studies. J Health Soc Behav 38: 21–37. doi:10.2307/2955359.
  2. Bopp M, Braun J, Gutzwiller F, Faeh D. (2012). Health risk or resource? Gradual and independent association between self-rated health and mortality persists over 30 years. PloS One 7: e30795. doi:10.1371/journal.pone.0030795
  3. Pietila¨ inen O, Laaksonen M, Rahkonen O, Lahelma E. (2011). Self-rated health as a predictor of disability retirement–the contribution of ill-health and working conditions. PloS One 6: e25004. doi:10.1371/journal.pone.0025004.
  4. Jylha¨ M. (2009). What is self-rated health and why does it predict mortality? Towards a unified conceptual model. Soc Sci Med 69: 307–316.
  5. Idler EL, Kasl SV. (1995). Self-ratings of health: do they also predict change in functional ability? J Gerontol B Psychol Sci Soc Sci 50: S344–S353.
  6. Hubbard RA, Inoue LYT, Diehr P. (2009). Joint modeling of self-rated health and changes in physical functioning. J Am Stat Assoc 104: 912. doi:10.1198/jasa.2009.ap08423.
  7. Wuorela M, Lavonius S, Salminen M, Vahlberg T, Viitanen M, Viikari L. (2020). Self-rated health and objective health status as predictors of all-cause mortality among older people: a prospective study with a 5-, 10-, and 27-year follow-up. BMC Geriatr. 20:120. org/10.1186/s12877-020-01516-9

Comment #6. There are many complex sentences that could be simplified by using full stops as opposed to semi-colon. An example is the sentence from lines 60 to 64; 85 to 88; 

â–ºResponse: According to the reviewer’s comment, we removed some semi-colons from long sentences and simplified complex statements (Line 69, 87, 140, 312)

Reviewer 3 Report

Comments and Suggestions for Authors

Overall, I found the paper to be well-researched and articulated. The methodology and results are robust and contribute significantly to the field. I noted a minor issue in the headlines of Table 1 where some corrections are needed for clarity, although it is explained in the text (166-167). Specifically, in the first part of the Table there are frequencies and proportion and in the second part there are means and standard deviation. In the text (233-238), describing Table 3 there is no explanation what Model 1 and 2 are. Addressing these minor points will enhance the overall presentation of the results.

Author Response

Comment #1. I noted a minor issue in the headlines of Table 1 where some corrections are needed for clarity, although it is explained in the text (166-167). Specifically, in the first part of the Table there are frequencies and proportion and in the second part there are means and standard deviation.

â–ºResponse: We inserted the names of subheader in the headline of Table 1.

Comment #2. In the text (233-238), describing Table 3 there is no explanation what Model 1 and 2 are. Addressing these minor points will enhance the overall presentation of the results.

â–ºResponse: The brief explanations on the Models have already been presented in Lines 231-235. However, to make it clearer, as the reviewer pointed out, we revised the sentence to read “Table 3 shows the results of multiple linear regression analysis conducted to assess the impact of interpersonal communication (Model 1) and instrumental use (Model 2) of internet devices on SRH and depressive symptoms after controlling for covariates (sex, age, marital status, living with children, religion, residency area, education, equivalent household income, and economic activity)”